# Bridging the Gap between Theory, Practice, and Policy: A Decision-Making Process Based on Public Health Evidence Feasible in Multi-Stage Research on Biological Risk Factors in Poland

**DOI:** 10.3390/ijerph17207657

**Published:** 2020-10-20

**Authors:** Anita Gębska-Kuczerowska, Sudakshina Lahiri, Robert Gajda

**Affiliations:** 1Faculty of Medicine, Collegium Medicum, Cardinal Stefan Wyszyński University, Kazimierza Wóycickiego 1/3, 01-938 Warsaw, Poland; 2National Institute of Public Health, Chocimska 24, 00-791 Warsaw, Poland; 3Institute of Digital Healthcare, WMG, University of Warwick, Coventry CV4 7AL, UK; S.Lahiri@warwick.ac.uk; 4Gajda-Med Medical Center, Piotra Skargi 23/29, 06-100 Pułtusk, Poland; gajda@gajdamed.pl

**Keywords:** continuous quality improvement, healthcare sector, biology risk control, patient safety, research to practice

## Abstract

Stakeholder input into the decision-making process when developing public health programs and policies is crucial. This article presents an innovative approach, involving online participation with a wide group of stakeholders located in different geographic locations for policy consensus. The results of the project have been used to propose assumptions regarding a strategy for preventing blood-borne diseases in Poland. The research was conducted iteratively using a multi-stage qualitative methodology to explore risk assessment involving blood-borne infections. The final output of the study is a list of key problems/challenges and potential solutions associated with medical and nonmedical services that are connected to the breakage of tissue continuity. Qualitative research is rare in risk assessment, as priority is generally given to statistical data and endpoints. In addition to policy preparation for blood-borne illnesses, the methodology employed in the study can also be used to successfully explore other areas of public health.

## 1. Introduction

The urgent need to bridge the gap between science, practice, and policy in public health has accelerated in recent years [1,2,3,4,5]. Often, policymakers’ decisions are a result of their goodwill and knowledge, along with social and economic pressures. It would be optimal to make decisions based on scientific circumstances and facts; however, in practice, this is not always feasible. An additional barrier to a conscious approach to policy-related decision-making in the field of healthcare is that the effects of actions are shifted in time, whereas cost considerations exist at very early, though critical, stages, such as preliminary analyses of information and strategy preparation. Therefore, convincing arguments, informed by theory and practice on the part of researchers, are needed in order to persuade legislators to make decisions—the effects of which are, at times, visible only after their tenure ends. Innovative program proposal requires a thorough analysis of each of the stages of its development, such that the obtained portrayal of a problem and the proposed process of change can be implemented. Indeed, strategies to implement evidence-based practices at a healthcare system level have been recently explored in terms of facilitators and barriers to the implementation process [6].

Information is a crucial element in decision-making [7,8,9]. When program proposals are prepared, attention should be paid to the element of participation by actively seeking inputs from those concerned with an analytical decision-making process through social consultancy to ensure that wider expectations have been taken into consideration. Therefore, as an argument to the critics of evidence-based public health, the intention of the current authors is to present a method that can potentially reduce the gap between science and policy-making through a systematic approach to problem-solving.

Many researchers and legislators believe that it is necessary to involve stakeholders into the decision-making process when developing programs and policies [10,11]. Likewise, a preliminary definition of the rules regarding the directions of reasonable decisions, related to the current knowledge about clinical sciences and public health, has also taken on tremendous importance [12,13,14,15]. In this regard, a recent review by Mathieson and colleagues uncovered a number of factors that serve as either barriers or facilitators to the use of theory and evidence when planning, guiding, and evaluating the implementation of policies and innovations [6]. The authors found that successful implementation results not only from a well-prepared plan (i.e., the what), but also from information on the process (i.e., the who, how) from previously obtained knowledge about barriers and facilitators. According to the authors, perceived barriers to implementation may result from fear of introducing innovations, which may initially seem destabilizing (for example, organizational restructuring that might be construed as either decentralization or centralization). Another barrier reported was an impaired organizational system, involving issues of hierarchy, leadership, management, and the flow of data and information. Equally important, they caution against proposing innovations without prior organizational preparation (i.e., infrastructure, people) as this might end up as a “falsestart” [6].

The objective of this article is to present a methodology for a multi-stage analysis process using a qualitative framework to prepare assumptions for a public health program strategy with wider inputs to gain policymakers’ support, especially with regard to prioritizing the goals of the program. The specific focus chosen was prevention of blood-borne illnesses. While medical progress has made it possible to diagnose and cure most blood-borne diseases, however, some of these conditions are not completely understood at present, making prevention especially important, including due to the economic costs involved [16,17,18].

The prevention of BBIs and related complications refer to a wide range and scope of activities in public health. In accordance with the epidemiological approach, it is important to take actions related to decreasing the number of infections in the population, reducing the chance and risk of transmission of infections, as well as harm reduction after exposure to the pathogen. In practice, the reduction in the number of sources of infection in the population concerns pathogens’ diagnosis, such as the detection of new cases and their effective treatment, increasing awareness of the risks and limiting new infections, and the microbiological safety of medical products and devices. Reduced chances of infections are a result of safe health behavior and preventative practices. Yet another component of prevention is awareness-building through health education, and increasing current understanding of threats and risks. Knowledge on potential sources of infection and elimination of threats, such as labeling and processing of infectious material and sharp equipment, is also of great importance. The reduction of the risk of infection transmission are can be achieved through the use of microbiologically safe equipment, along with compliance with procedures and regular epidemiological supervision aimed at ensuring the quality and safety of high-risk services. It is equally important to update the list of sources and routes of infection by monitoring high-risk areas. Harm reduction post-pathogen exposure is associated with active prevention (vaccination, such as HBV) and an efficient system for the registration of all cases of exposure and their immediate and effective treatment [19,20].

## 2. Materials and Methods

The current research was proceeded by a pilot study [21,22] which included 16 regional epidemiology consultants representing all voivodeships, that is, health administrative regions, at the national level in Poland. Approval for the research was received from the Research Board (steering committee of the Project KIK 35) of the National Institute of Public Health (NIPH). The Institute was established almost a century ago as the National Institute of Hygiene with disease prevention as one its key goals. The study protocol ensured voluntary participation and anonymity of all participants. Informed consent was obtained before the study from all participants.

The study was conducted in three stages, as illustrated in Figure 1. These stages helped with the identification of the problem, process, and proposed solutions, as well as task prioritization.

### 2.1. Sample

Stakeholders were defined as individuals who had a substantial role in advancing the development of a blood-borne infection (BBI) prevention program. Stakeholder opinions were obtained from a broad group of experts and practitioners located throughout the country. A total of 111 participants, representing all 16 Polish health administrative regions, took part in the study. Three groups of experts were invited to participate: practitioners; employees of supervising authorities (administrative supervision) at the regional level; policy-makers; and opinion leaders, that is, decision-makers, at the national level. The study was carried out in the form of a two-panel internet forum discussion (quasi-FGI), and a focus group (FGI).

The first two groups participated via forums that were conducted using IDEABLOG^®^ from Kantar Mill Word Brown, London, UK (first three days as individual input and the remaining two days as common forum), an independent online interviewing platform, whereas the third group of stakeholders, that is, experts and administrative supervisors, provided inputs through in-person focus group interviews (FGI). The use of an online platform was deemed important as it provided participants with a degree of flexibility to express their views and at a pace that was manageable with their individual schedules. The participation through an independent interviewing company and the anonymity of experts helped to reduce the risk of biases in the views expressed.

Participants in Stage 1 comprised of management staff and practitioners, specifically epidemiologists, experts, and service providers whose daily work was associated with the risk of BBIs, both exposure to the risk and risk generation. The perspective of an authority that oversaw the observance of regulations regarding the minimization of risks, that is, the Sanitary Inspectorate, was also obtained at this stage. Participants were invited using a list prepared by the NIPH. The official invitation contained a presentation of the research project, its objectives, and a request to participate in the study. Due to the novel nature of the vehicle of their participation, that is, online forums, it was important to over-recruit in order to ensure a suitable response rate. To date, little is known about adequate response rates involving online forums. However, studies involving focus groups suggest a minimum of five participants as acceptable and this benchmark served as a guide for Stage 1. Hence, for this stage, 80 individuals were contacted, of which 42 agreed to take part in the first online forum (Forum 1) [23].

For Stage 2, representatives of supervising agencies, such as state administration authorities, medical center managers, experts from academic centers, and specialists and experts from the field of public health were contacted. Of the 125 people invited to participate, 41 took part in the study and formed the cohort for the second online forum, that is, Forum 2. The rationale for conducting the aforesaid fora was underpinned by the need to obtain information, both from the level of service implementation (Stage 1) as well as the management level (Stage 2). The main reason for refusal to participate in the study at each stage was an excessive number of other obligations, which participants indicated would have made it difficult for them to be completely involved in the project.

Lastly, Stage 3 consisted of one set of FGI. Generally, FGIs are a qualitative method used to discuss particular themes with a group of invited participants in an open atmosphere with proposed scenarios [24,25]. In total, 11 opinion leaders and administrative supervisors were invited, all of whom participated in the study. Table 1 provides a detailed outline of participants.

### 2.2. Data Collection

In-depth views and opinions were obtained focusing on two key areas: (1) The problem of reducing the risk of infections in services and procedures connected with abrasion and damage of tissues; and (2) risks associated with the spread of infections and protection of patients, staff, and clients. The exploratory nature of the study lent well to a qualitative research framework, as it allowed for understanding of the decisions made by participants regarding a particular policy stance and factors underpinning the decision-making [26,27]. Table 2 depicts the topic guide that was used with each of the three groups of stakeholders to collect the necessary data.

Informed consent was obtained from all groups of participants prior to the actual proceedings. There was a note-taker, and independent observers supervised the online quasi-FGIs and FGI.

The first stage of this research project (i.e., online Forum 1) was carried out between 18 July and 20 August 2016. Throughout the forum, the invited epidemiologists and experts were given two topics every day, on which they were asked for their views and opinions. On each topic, the necessity to support answers with evidence was emphasized; for example, from practice, figures, statistical data, along with particular examples reflecting problems and good practice. The presented information was verified by participants through their own publications, practice case descriptions, and synthetic expert opinion. These elements constituted the quantifiable information or source data (SD) provided by the participants. The first three days involved conducting the individual interviews. The last two days consisted of a joint discussion about the information that was collected until then. During the joint discussion, each participant only had access to general statements, but not the SD.

The data collected were verified through the SD, and these constituted an extensive collection of research material that was arranged in a strengths, weaknesses, opportunities, and threats (SWOT) analysis which was then used for the subsequent second stage. The SWOT analysis allowed for organization of the first-stage data in a detailed report, based on suggestions that were made by participants during the joint discussion. The forum and associated discussion covered all aspects of BBI risk analysis. The information obtained was available for review by all participants in the final stage of the study (FGI). At the same time, all items were assessed by strengths or weaknesses, as well as how participants saw opportunities and limitations from their professional perspective.

Information for Stage 2 was gathered via the online Forum 2 and also conducted as a quasi-FGI between 5 September and 7 November 2016. In a similar methodology as Stage 1, participants received one topic per day for five days, and were asked to express their views and opinions regarding the topics. Figures and data collected and verified (from Stages 1 and 2) constituted the basis for the subsequent analysis. During the second forum, the first-stage data were consulted while also obtaining new data. The key output from Stage 2 was a list of 33 proposed solutions or postulates, which participants were asked to categorize based on groupings that they viewed as appropriate. This led to categorizing the postulates into the following topics: management, legislation, education, and financing, which is compatible with the administrative structure of healthcare management. Proposals for the 33 postulates formed the basis of discussion for the third stage of the project. As part of preparation for the FGI, the 33 proposed solutions from Stage 2 were further reviewed by research staff and reclassified into four categories: legislative, educational, organizational, and financial. These were emailed to the FGI invitees prior to the meeting so that they could adequately prepare for the discussion.

Finally, the in-person FGI that summed up the first and second stages of this study was completed on 30 November 2016 at the NIPH in Warsaw. The meeting was attended by opinion leaders and policy-makers who represented patients’ interests, service providers, public payers, the Agency for Health Technology Assessment and Tarification, and the Ministry of Health. The focus group lasted for about 90 min. Welcoming comments and presentation of the project details were followed by a general discussion on the subject of prophylaxis and prevention of BBIs. Next, a list of priorities, selected during the previous two stages of the study was introduced, along with a discussion exploring approximate timelines that would be needed to achieve any chosen priorities. Such an arrangement was aimed at redirecting specific tasks to the listed areas of national policy and administration. Participants were encouraged to freely discuss all topics that came up during the proceedings. The proceedings were overseen by a moderator whose role was to create conditions favorable for discussion.

At the FGI, participants were asked to prioritize the 33 proposed solutions. Then, they assigned priority in terms of time and urgency of implementation, according to the principle of the highest scoring. Rough rank ordering was used for the prioritization process. For example, one task could get a maximum of three points. In this way, the postulates were assigned a rank. In the first group of the most important priorities (i.e., high number of credits 11–18), four postulates were included. In the second group of moderate priorities (4–7 credits), there were seven postulates. Finally, in the third group of low importance (1–3 credits), 15 postulates were included and seven were rejected. In addition, under prioritization of tasks, similar to the earlier stages, the adopted topic areas were classified into legislation, education, organization, and finance, which are recognized areas from the policy perspective and arrangement of administration. The final outcome of this study was the rank-ordered 33 postulates (see Amendment).

### 2.3. Data Analysis: Quality and Verification

A broad thematic approach was applied to the gathered data. Research staff simultaneously reviewed all evidence and SD and proceedings of the two online forums and the FGI (as well as the tracks of interviews). The proposed data-ordering at Stages 1 and 2 according to the SWOT analysis method and division according to the proposed criteria were carried out jointly by the respondents and experts from their professional perspective. The research team met to discuss the broad themes that arose from the proceedings, and refined these to arrive at the postulates and their categorization under the four areas, that is, legislation, education, organization, and finance.

## 3. Results

Stage 1: Discussions from this stage led to the identification of the following risk factors for BBIs by the participating epidemiologists—namely, HBC, HCV, and HIV. The risk factors and pathways of BBI dissemination were recognized in accordance with scientific evidence. In the context of this study, the BBI population risk was categorized by participants and based on statistics and indicators (incidence, prevalence, medical treatment of BBIs, and associated complications of BBIs). Additionally, the risks to professionals and providers were also identified, particularly involving procedure types and injuries during procedures, as well as waste disposal of therapeutic materials. For example, it was identified that nurses were more at risk of exposure to BBI-related adverse events, and that most of these events were caused by self-injuries, such as injections and cleaning of devices. Moreover, the importance of keeping an audit of occupational risk factors related to the afore mentioned factors was also discussed, as was medical tracking.

*Exposure to hepatitis infections*. Experts also identified a level of awareness of BBIs among professionals in different service sectors, with a related high risk of BBI. Participants emphasized that staff must follow both proper procedures related to handling BBI pathogens, and also follow preventative measures. Moreover, outsourcing of cleaning equipment and devices was also seen as a risk of spreading BBIs. Staff should also be mindful that procedures for preventing BBIs are followed according to best-practice protocol. Based on the above discussions, the epidemiologists concluded that there was a rather stable situation in Poland in selected diseases from the group of BBIs (Appendix A). The results on the risks of BBI analyses were grouped into areas indicating the strengths and weaknesses of risk management and the opportunities and threats leading to the SWOT analysis which formed the input to Stage 2.

Stage 2: Initially, participants in this stage were invited to present their experiences and knowledge regarding information collected from Stage 1. Additionally, they were also asked to weigh in on program management concerns regarding BBI within their localities and issues of policy overseeing BBI in the context of infectious diseases generally. Participants also indicated the area of activity for the policy to resolve the presented BBI issues from Stage 1. The main output from this stage was the formulation of the 33 postulates which were assigned to the following areas: management, legislation, education, and financing.

Stage 3: The key output from this stage was the rank ordering of the 33 postulates from the perspectives of feasibility and public health program capacity.

## 4. Discussion

Taking a “bottom-up” data collection approach to the presented methodology, we selected epidemiological diagnosis and risk analysis evidence from the perspective of service providers, and then a decision-making process analysis. The next step entailed a presentation of both the problems and their management, and finally, rank ordering of the collected findings.

Due to the practical nature of the current research, a qualitative method that enabled the collection of different opinions, evidence, and postulates from a wide group of stakeholders was selected. This approach is in line with the new method of managing public health proposals, as qualitative methods provide participants with a space for free expression of thoughts, while also being evidence-based [28]. The techniques and web tools employed in qualitative research also provided a space for discussion and exchange of opinions, such as time for consultations, locating evidence, and protecting anonymity. In turn, this allowed for the obtaining of responses from participants, reflecting not only the direction of their views (e.g., what they find to be positive or negative), but also their substantiation (e.g., explanations for their thoughts), supported by examples and evidence. Therefore, the application of free-discussion techniques, such as the use of an online platform, was necessary.

Unlike standard interview techniques, the online platform enabled a flexible reaction to participants’ responses and the expansion of resultant information. In the field of public health, qualitative techniques comprise an additional, valuable source of information that enables the rationalization of a decision-making process by providing an opportunity to take a broader view of the problem, and includes other stakeholders in the interpretation and decision-making process [29,30,31,32,33]. To that end, the use of the online platform IDEABLOG^®^ turned out to be a useful tool in the context of this study.

The evaluation of the reliability and quality of data analyses is usually based on quantitative approaches. For example, known methodologies, such as GRADE and AGREE, are largely related to the implementation of results from other studies into policy [34,35,36,37]. In the analysis of data, especially in relation to the assessment of efficacy and safety of treatment, these studies are of indisputable importance. Relatively less importance has been attached to qualitative studies, which also require different methodologies and indications for their application.

The methodology of the multi-stage process of research and analysis as presented in this article involved epidemiology experts who helped with the identification of BBI risk factors and risk analysis underpinned by evidence through their repository of things such as statistical data, research results, expert opinions, case studies, and publications, which were a condition for participating in the first and second stages. The project included a sequence of diagnoses of the situation, results from the first stage and diagnosis of decision-making processes ensuing from the second stage, and a discussion on a coherent catalogue of the developed data of the first and second stages, which then led to results of the third stage. References to practical links between theory and practice were emphasized both at the stage of problem diagnosis and at that of programs/procedures. Such an approach is in step with the opinions of many other researchers, and includes both the creation of recommendations and their updating based on data [33,34,35,36,37].

The present analysis, divided into status diagnoses (Stages 1 and 2) and decision-making processes (Stage 3) was an attempt to systemize information about the extent of the risks associated with BBI and the essential stages of analysis for the preparation of further stages (i.e., proposed solutions). This is consistent with a methodological “mixed research approach” represented by other researchers in public health and healthcare policy [38,39,40,41,42]. Through its use of qualitative inquiry, the current study aimed to understand the extent to which the identified problems and opinions are common among the group of participating experts. However, respondents were asked for their opinions and to provide supporting evidence in the form of statistical data and other sources.

First, the study collected qualitative data and figures and diagnosed the status of knowledge (practical and theoretical) and the functioning of the system of services with regard to a problem researched. The web tool, IDEABLOG^®^, was especially helpful to widening participation, as study participants were located in different parts of the country. In the final stage, the collected data on the diagnosis of the system and assessment of the scale of the risk of BBIs was systematized, and priorities were identified through rank ordering.

In terms of public health evidence, the results of this research, in the form of extensive reports from each research stage, were submitted to the Hepatitis C-virus Project Board (KIK35) to propose a new strategy for a policy of BBI prevention. These reports were also presented to all stakeholders as a source of information on the issue of blood-borne diseases in Poland. The selection of solutions resulted from substantive reasons, that is, the intention to collect possibly the most extensive material about the foundations and the actions taken by various institutions in the context of the prevention of BBIs, as well as organizational and technical reasons, since the target groups presented all health regions in Poland and people with different professional backgrounds and responsibilities.. The dissemination of the findings of the analysis to the stakeholders resulted in a greater degree of awareness and legislative changes; for example, of a greater sanitary regime in non-medical services and educational changes related to the inclusion of a range of issues in vocational education programs. To that end, some results were incorporated into policies and programs by the Ministry of Health, the Chief Sanitary Inspectorate, National Training Centre for Nurses and Midwives, and National Institute of Public Health.

Particularly in this study, the discussions focused on biological risk factors, but the methodology employed in our research might be useful to explore other risk factors. In the first stage, the risk of infection in services was assessed. In the second stage, infection risk issues related to processes and procedures were examined. In the third stage, decision-makers were asked to prioritize the lessons learned in the previous stages. Economic realities and the dynamics of changes in the field of epidemiology, as well as pressure from patients, clients, and service providers who are aware of the risk, provide the possibility of numerous decisions. This increases the obligation of experts to prepare recommendations for public health programs and policies considering broader insights around the particular problem.

This study also had certain limitations. The detailed analysis did not consider current regulations or organizational and financing structures; instead, it only indirectly referred to the opinions of experts and the information provided by them related to these elements. Moreover, some of the invited experts were unable to participate due to their professional duties, while others felt that there might be a possibility of political pressure.

Nevertheless, the findings of the present study provide directions for further research. In the future, the current authors will verify the extent to which the present results contributed to improvements in the field, for example, health conditions related to BBIs and whether the presented method of priority selection will find support. Statistical data on health risks allows for assessment of health implications, while the applied methodology also allowed for diagnosis of the processes leading to these effects. A limitation of the qualitative method is that it is not possible to quantify directly the risk, which has made it possible to look at the infection problem from a different perspective. However, the choice of the presented method was dictated by the necessity of the chosen objective. Acknowledging the limitations, the study recommended a method wherein experts were asked to support their views and decisions with analysis of hard data and research results from their own repository.

## 5. Conclusions

Evidence-based public health is routine. There is also an ongoing need to seek methods for evaluating the effects and processes of action taken in public health management. A standard sentinel system and epidemiological research provide strong statistical evidence and are the basis for rational decision-making. Improvement of the quality of statistical data and the need to broaden knowledge with new and better diagnostic tools inspire the search for better qualitative and quantitative methods in public health. This is particularly important in areas of public health in which there are similar risks and different approaches in terms of prevention, as well as different regulatory responsibilities.

This study examined some of the aforesaid principles through the lens of the challenges involving the problem of blood-borne diseases from the perspectives of medical and non-medical services, particularly to see how stakeholders approached issues of prevention surrounding BBIs in the services sector.

The methodology can be useful for the development of a strategy for the prevention of BBIs, but also for other risk factors. The methodology applied and the results obtained also inspired changes in education programs for employees in both the medical and non-medical service sectors in the prevention of BBIs. This allowed for the consolidation of various professional and social environments in order to reach a common consensus regarding the development of an infection risk-reduction program.

The report from the three-stage analysis was presented for decision-makers at various levels of management (in the medical and non-medical service sectors). As an outcome of the study, a proposal for a health policy prevention strategy for BBIs was then prepared with input from the research. The policy included extension of education programs based on analysis and educational material in direct e-learning. Additionally, specialized training programs for nurses were amended, indicating the need to modify the training programs for other professional groups.

## Figures and Tables

**Figure 1 ijerph-17-07657-f001:**
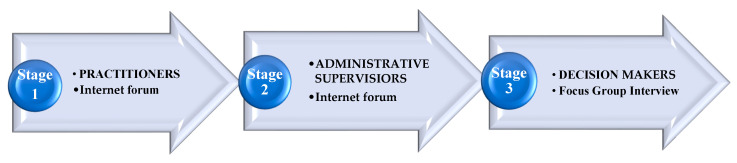
Project scheme—research project stages.

**Table 1 ijerph-17-07657-t001:** Profile of interviewees and knowledge supporters.

Descriptor	Breakdown ^1^
Online Forum 1 participants (n = 42) (epidemiology experts, epidemiology administrative supervisors)	Practitioners and staff who had everyday contact with the subject of supervision, prevention, and combating BBIs. Representation covered 100% of the Polish health administrative regions.
Online Forum 2 participants (n = 41) (public health experts, healthcare administrative supervisors)	Representatives of supervisory institutions; state administration offices, medical centers, and academic units (public health specialists). Representation covered 88% of the Polish health administrative regions.
Focus Group Interviews participants (n = 11) of opinion leaders, policy-makers, decision-makers	Organizations represented: Ministry of Health; Agency for Health Technology Assessment and Pricing; National Health Fund; Country Public Health Consultant; Chief Sanitary Inspector of the Country; NGO (patient representatives and clients); Supreme Medical Chamber; Supreme Chamber of Nurses & Midwives; representatives of medical university and National Institute of Public Health-National Institute of Hygiene

^1^ An individual breakdown is not provided to protect participant anonymity.

**Table 2 ijerph-17-07657-t002:** Topic guide used with specific participant groups.

**Online Interview and Forum 1: Practitioners (Epidemiology Field) ^1^**
Role of participant in BBIs ^2^ in daily work Level of awareness of BBIs Provisions to reduce the risk of BBIs Practical implementation of the guidelines Reporting incidents of infections of staff and patients in medical and non-medical sectors Good practices in everyday work routine Education and training on BBIs Role of finance in preventing BBIs Importance of the disinfection process in reducing BBIs Awareness of the risk of BBIs among patients and clients
**Online interview and Forum 2: Management staff (public health field) ^1^**
Prevention of BBIs in respondent’s respective region Awareness and knowledge about BBIs Practical implementation of regulations concerning prevention of BBIs Infrastructure and the role of the employer and the state Conducting control
**Opinion leaders, administrative supervisors**
Prophylaxis and prevention of BBIs Goals and priorities setting on legislation, organization, education, and financing Time plans towards achieving the set goals

^1^ The topics were discussed in the open forum and users could see each other’s statements, but all usernames were pseudonymized. ^2^ Blood-borne infections.

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
