# Peer review of "Bridging the Gap between Theory, Practice, and Policy: A Decision-Making Process Based on Public Health Evidence Feasible in Multi-Stage Research on Biological Risk Factors in Poland"

_ijerph, 2020, doi:10.3390/ijerph17207657_

Round 1
Reviewer 1 Report
The suggested revisions were uptaded and now the manuscript is a good level for the publication.
This manuscript is a resubmission of an earlier submission. The following is a list of the peer review reports and author responses from that submission.
Round 1
Reviewer 1 Report
As this manuscript deals with the assessment of different risk factors. I think authors need to include a critical discussion about the following issue:
the risk factors analysed in this study how they can be categorized depending on the severity of the risks involved?
How the severity of risks impacting the decision making process?
Reviewer 2 Report
This is an interesting study, howover the tone of the conclusions is still overly strong. Indeed, the study does provide evidence as described, but the way the authors state this is too strong and does no is supported the statistical data. Both the abstract and the Discussion opening paragraph are worded very strongly and imply that this study is the last word on this argument.
The work has some statistical limitations. It would be necessary for the authors to add a part relating to statistical analysis of the data or to justify this limitation.
Reviewer 3 Report
Dear Authors,
Many congratulations on the successful completion and submission of your manuscript.
This paper presents a unique approach that involves a sequential multi-stage analysis process performed using qualitative as well as quantitative data, which explores the risk assessment involving blood-borne infections. This study primarily focused on the biological risk factors of BBIs, however, the authors suggest that such an approach can effectively be applied to address other potential risk factors as well thereby potentiating the development of a stratergy for the prevention of BBIs.
This study greatly emphasizes the need to bridge the gap between science and policymaking through a systematic approach and problem-solving, which is highly crucial in strengthening the decision-making process, for the analysis of related problems to gain the policymakers' support.
The authors may take note of the following minor changes that can be incurred in the manuscript.
Topic, Line 4: can change 'multistage' to "multi-stage"
Topic, Line 4: the term 'biology risk factors' authors can consider modifying it to 'biological risk factors'
Line 77: can modify to something like this '...important, considering the economic costs involved'
Figure 1: If possible the schematic diagram can be replaced with an imag with higher resolution
Line 200: replace 'in the' to 'under prioritization...'
Line 285: can consider replacing 'biology risk factors' to 'biological risk factors'
Thank you!
